# Gene Variants Related to Cardiovascular and Pulmonary Diseases May Correlate with Severe Outcome of COVID-19

**DOI:** 10.3390/ijms23158696

**Published:** 2022-08-04

**Authors:** Mateusz Sypniewski, Zbigniew J. Król, Joanna Szyda, Elżbieta Kaja, Magdalena Mroczek, Tomasz Suchocki, Adrian Lejman, Maria Stępień, Piotr Topolski, Maciej Dąbrowski, Krzysztof Kotlarz, Angelika Aplas, Michał Wasiak, Marzena Wojtaszewska, Paweł Zawadzki, Agnieszka Pawlak, Robert Gil, Paula Dobosz, Joanna Stojak

**Affiliations:** 1Department of Genetics and Animal Breeding, Poznan University of Life Sciences, 60-637 Poznan, Poland; 2Central Clinical Hospital of Ministry of the Interior and Administration in Warsaw, 02-507 Warsaw, Poland; 3Biostatistics Group, Department of Genetics, Wrocław University of Environmental and Life Sciences, 51-631 Wrocław, Poland; 4National Research Institute of Animal Production, Krakowska 1, 32-083 Balice, Poland; 5Department of Medical Chemistry and Laboratory Medicine, Poznan University of Medical Sciences, 60-806 Poznan, Poland; 6Center for Cardiovascular Genetics & Gene Diagnostics, Foundation for People with Rare Diseases, 8952 Schlieren-Zurich, Switzerland; 7MNM Bioscience Inc., 1 Broadway, Cambridge, MA 02142, USA; 8Department of Infectious Diseases, Doctoral School, Medical University of Lublin, 20-059 Lublin, Poland; 9Department of Hematology, Frederic Chopin Provincial Teaching Hospital No. 1 in Rzeszow, 35-055 Rzeszow, Poland

**Keywords:** cardiovascular diseases, pulmonary diseases, genetic variants, risk factors, COVID-19, GWAS, single nucleotide polymorphism

## Abstract

**Background:** Severe outcomes of COVID-19 account for up to 15% of all cases. The study aims to check if any gene variants related to cardiovascular (CVD) and pulmonary diseases (PD) are correlated with a severe outcome of COVID-19 in a Polish cohort of COVID-19 patients. **Methods:** In this study, a subset of 747 samples from unrelated individuals collected across Poland in 2020 and 2021 was used and whole-genome sequencing was performed. **Results:** The GWAS analysis of SNPs and short indels located in genes related to CVD identified one variant significant in COVID-19 severe outcome in the *HADHA* gene, while for the PD gene panel, we found two significant variants in the *DRC1* gene. In this study, both potentially protective and risk variants were identified, of which variants in the *HADHA* gene deserve the most attention. **Conclusions:** This is the first study reporting the association between the *HADHA* and *DRC1* genetic variants and COVID-19 severe outcome based on the cohort WGS analysis. Although all the identified variants are localised in introns, they may be correlated and therefore inherited along with other risk variants, potentially causative to severe outcome of COVID-19 but not discovered yet.

## 1. Introduction

Between 1 January 2020 and 31 December 2021, the pandemic of coronavirus disease 2019 (COVID-19) caused a global excess mortality of 14.91 million people [1]. Despite the availability of vaccines against this disease, it is still a critical issue in public health. Among the most important sociodemographic and personal risk factors of COVID-19 are age, male sex, and comorbidities [2,3,4], but these do not fully explain different individual responses to this disease and its different outcomes.

Severe outcomes of COVID-19 account for up to 15% of all cases [5] and lead to life-threatening complications, such as acute respiratory distress syndrome; acute cardiac, kidney, and liver injury; thromboembolic events; septic shock; and multiorgan failure. Previous studies reported several pre-existing medical conditions that might accelerate the progression of COVID-19 and be a risk factor for the severe outcome of the disease. Most of the conditions are related to cardiovascular disorders, mainly cardiovascular (CVD) and pulmonary diseases (PD) [2,3,4]. For instance, a cohort study from China showed that the fatality rate among patients with CVD was 10.5% [6]. Moreover, the epidemiological study and meta-analyses indicated that cardiovascular diseases (such as heart failure or chronic stable heart disease) were significant risk factors for severe COVID-19 [7,8]. Studies on the influence of respiratory diseases, such as asthma and chronic obstructive lung disease, showed more inconsistencies in the impact of pre-existing PD on the severe COVID-19 course [8,9].

Nowadays, cardiovascular diseases are a leading cause of death in the world and their incidence rises sharply with age. However, not only socio-economic factors proved to be a risk for CVD, but there is also a genetic component that may predispose to the CVD and be a risk factor for a severe COVID-19 course. The study on the association of 16 polymorphisms in genes encoding prothrombotic and cardiovascular risk factors with COVID-19 disease severity demonstrated that integrin beta-3 (*ITGB3*) PIA1/A2 polymorphism was independently associated with the increased risk for severe COVID-19. Moreover, the single nucleotide polymorphism in the β-fibrinogen gene was detected in the homozygous mutated form only in patients that suffered from severe COVID-19 [10]. All these data suggest that there may be a genetic factor contributing to the severe COVID-19 course that is also associated with genes related to CVD. In the case of PD, the association between genetic variants and a severe COVID-19 course was less consistent. A meta-analysis indicated that patients with comorbid chronic obstructive pulmonary diseases (COPD) and chronic respiratory diseases were more susceptible to severe COVID-19, while no association between asthma and severe COVID-19 was identified [11]. Acevedo et al. [12] reported that patients suffering from COPD had increased levels of 41 plasma proteins, mainly related to the inflammatory response, which is also characteristic for patients with severe COVID-19 infection. Although some studies identified a positive correlation between some pulmonary conditions, such as COPD and severe COVID-19 on the clinical level [11], the genetic basis of this correlation could not be identified, even in the large studies including populations of various ethnicities [13]. Previous studies reported that patients suffering from CVD are at higher risk of developing a severe course of COVID-19 with smaller chances of recovery [7]. 

Therefore, the aim of this study was to test whether genetic variants located in genes, previously described in the literature as related to cardiovascular and pulmonary diseases, are also associated with a severe course of COVID-19 in the Polish cohort of 232 patients who suffered from severe, life-threatening cases of COVID-19 infection. 

## 2. Results

### 2.1. Variants Related to the Cardiovascular Panel and the Severe Outcome of COVID-19

We have analysed 19,130 SNPs and indels located in genes related to CVD but none of the tested variants passed the set significance thresholds. The variant with the lowest *p*-value was located in the Hydroxyacyl-CoA Dehydrogenase Trifunctional Multienzyme Complex Subunit Alpha (*HADHA*) gene (OR = 0.59, raw *p*-value $1.4\times 10^{-6}$) (Figure 1, Table 1). This variant was located on chromosome 2, in intron, and was determined as a potential protective variant in regard to the severe course of COVID-19 infection. 

### 2.2. Variants Related to Pulmonary Panel and Severe Outcome of COVID-19

The logistic regression on 7423 SNPs and short indels for genes related to pulmonary diseases identified two variants significant in COVID-19 severe outcome, in Dynein Regulatory Complex Subunit 1 (*DRC1*) with ORs of 0.56 and 1.71 (Figure 2, Table 2). These two variants were located on chromosome 2, in introns, and the first of them was determined as a potential protective variant, while the second was determined as a risk variant in regard to the severe course of COVID-19 infection. 

## 3. Discussion

After more than two years of the battle with the SARS-CoV-2 coronavirus, it is clear that CVDs and PDs are significant risk factors for a severe COVID-19 outcome. Especially cardiac disease, such as heart failure, and coronary artery disease predispose to a severe COVID-19 course [7,8]. Pulmonary disorders, such as COPD, are associated with higher COVID-19 mortality [14]. We investigated whether these clinical associations correspond to genetic variants related to CVD or PD. In our study, we searched for variants significantly associated with COVID-19 severity that have clinical implications within gene panels related to CVD and PD. 

In contrast to the majority of studies that mainly reported SNPs with an alternative allele increasing the risk of a severe COVID-19 infection outcome, in the case of the cardiovascular disease gene panel, we identified a potentially protective variant located in the *HADHA* gene with an OR of 0.59 (Table 1). The *HADHA* gene is involved in the production of the mitochondrial trifunctional protein (MTP) that contains three enzymes essential for fatty acid oxidation. *HADHA* is required for fatty acid beta-oxidation and cardiolipin remodelling, essential for mitochondrial functions in human cardiomyocytes, and a mutation in this gene leads to long-chain 3-hydroxyacyl-coa-dehydrogenase deficiency [15,16,17]. The impaired 3-hydroxyacyl-coa-dehydrogenase metabolism may lead to hypoglycaemia and tissue damage of the heart, liver, and muscles [18]. We hypothesise that the identified protective variant localised in the *HADHA* gene improves fatty acid metabolism which fosters a COVID-19 mild outcome.

Interestingly, one study showed that mutations in the HADHA gene should be considered as a risk factor for a severe outcome of COVID-19. The patient, described in a single case report, with pathogenic variants in HADHA, suddenly developed acute respiratory failure due to COVID-19 infection, accompanied by refractory hypotension from severe cardiomyopathy which led to multiple organ failure and death [19].

Considering genes related to pulmonary diseases (PD), we identified two variants in the *DRC1* gene significantly associated with COVID-19 severe outcome. Previous studies from Japan showed that variation (large homozygous deletion) in *DRC1* plays an important role in primary ciliary dyskinesia (PCD), a rare genetic disorder that prevents the clearance of mucous from the lungs, leading to frequent respiratory infections caused by bacteria and other irritants in the mucous [20,21,22]. Moreover, the *DRC1*/*CCDC164* mutation significant in PCD was identified in India [23]. To the best of our knowledge, our study is the first to describe the association between the variation in DRC1 and COVID-19.

Since the beginning of the COVID-19 pandemic, patients with pre-existing CVD and PD, due to significantly worse outcomes of the disease, have been treated as of the highest risk of severe COVID-19. The presented study helped to narrow down the list of genes involved in cardiovascular and pulmonary diseases which may be important in the COVID-19 infection course. All the identified variants are located in introns, which explains why they were not detected in SNP array and whole-exome sequencing studies (WES) [24]. Due to their genomic location in non-coding regions, these variants are not expected to be causal mutations themselves. Although the obtained results were based on the Polish population exclusively, we assume that they can be regarded as representative for Central European populations [25]. The identified small variants should be considered as novel candidates in the prediction of a COVID-19 infection outcome. Further WGS-based research involving a much larger cohort is necessary to verify the candidates and identify the associated causal mutations. 

## 4. Materials and Methods

### 4.1. Sample Collection

In this study, a subset of 747 samples from unrelated individuals collected across Poland between April 2020 and April 2021 was used. Only individuals of Polish origin and without diagnosed severe genetic disorders (up to the moment of sample collection) were qualified for this study. Within this cohort, the case group was SEVERE group (N = 232) composed of patients with severe, life-threatening cases of COVID-19 infection, including respiratory insufficiency, requiring intensive medical care and artificial ventilation resulting in the NEWS classification above 5. The remainder of the samples (N = 515), including patients with mild (N = 285) and asymptomatic (N = 230) outcomes, were treated as a control group. Detailed information about the cohort, including demographic and clinical features, can be found in Appendix A and in our previous paper by Kaja et al. [25].

### 4.2. Ethical Policy

All participants of this study provided informed consent before collection of blood samples and aforementioned basic clinical data. The ethical approval was granted by the Ethics Committee of the Central Clinical Hospital of the Ministry of Interior and Administration in Warsaw (decisions nos: 41/2020 from 3 April 2020 and 125/2020 from 1 July 2020). The study complied with the 1964 Helsinki declaration and its later amendments and adhered to the highest data security standards 140 of ISO 27001 and the General Data Protection Regulation (GDPR) act.

### 4.3. Total Quality Management

The project was carried out in accordance with the Total Quality Management (TQM) methodology, which ensures the quality of results and analyses the risk and possible difficulties. TQM requires defining all critical points of the procedures: reference ranges for collected biological material, its preparation, isolation, DNA concentration and quality, genomic sequencing, and quality control of the data. The legal and ethical transparency of the entire project was ensured, including confidentiality, integrity, and impartiality of the data.

### 4.4. Whole Genome Sequencing

The whole genomes of 747 unrelated participants were sequenced in this study. A total of 4 mL of K-EDTA peripheral blood from participants were collected according to a standardised Quality Management System protocol. Genomic DNA was isolated from the peripheral blood leukocytes using a QIAamp DNA Blood Mini Kit, Blood/Cell DNA Mini Kit (Syngen Biotech Sp. z o.o. Sp.K., Wroclaw, Poland) and Xpure Blood Kit (A&A Biotechnology, Gdansk, Poland) according to the manufacturer’s protocols. The concentration and purity of isolated DNA were measured using the NanoDropTM spectrophotometer, and the quality of the DNA was evaluated using gel electrophoresis. The sequencing library was prepared by Macrogen Europe (Amsterdam, The Netherlands), using TruSeq DNA PCR-free kit (Illumina Inc., San Diego, CA, USA) and 550 bp inserts. The quality of DNA libraries was measured using 2100 Bioanalyzer, Agilent Technologies, Santa Clara, CA, USA. The whole genome sequencing (WGS) was performed on the Illumina NovaSeq 6000 platform using 150 bp paired-end reads, yielding a mean depth of coverage of 35.26× in the cohort. Sequenced reads were mapped to GRCh38 human reference genome and variants were annotated using Ensembl Variant Effect Predictor v.97 [26]. More detailed information regarding data processing can be found in our previous paper by Kaja et al. [25].

### 4.5. Panels of Genes

For this study, we used the commercially available cardiology and pulmonology panels associated with cardiovascular and pulmonary disorders. The cardiovascular panel consisted of 205 genes with 10 kb buffer at both ends and 274,921 SNPs and indels (Appendix A). The pulmonary panel consisted of 131 genes with 10 kb buffer and 113,200 SNPs and indels (Appendix A). Both panels were collected from commercially available cardiology and pulmonology panels freely available online [27].

### 4.6. Tag Variants

Whole genome sequencing of the 747 genomes led to the identification of 38,296,203 SNPs and indels. In order to narrow down the search of risk and protective variants for further analysis from the full data set, we extracted variants corresponding to the cardiovascular and the pulmonary panels defined above. In the next step, variants were filtered for minor allele frequency (MAF) > 0.01, Hardy–Weinberg equilibrium at *p*-value < 10 × 10^−6^. For further narrowing the number of variants, we applied the so-called tag variant approach. In this method, instead of considering all available variants, we analysed only a subset of variants that were representative for haplotype blocks. To select the representative variants, we employed linkage disequilibrium pruning. In this approach, we considered a window of 50 variants, calculated LD between each pair of variants in the window and removed one of a pair of variants if r2 > 0.05, where r2 is square of the correlation coefficient between the presence or absence of an allele at first locus and second locus [28]. Pruning allows for the elimination of variants remaining in high LD and thus leaving only those which are representative for the whole haplotype blocks, i.e., tag variants, and reduction in type-I error rate by decreasing the number of tested variants. Both data filtering and variant pruning were implemented in PLINK v.1.9 [29].

### 4.7. Genome-Wide Association Study

For the estimation of the risk of development of severe symptoms of COVID-19 infection, genome-wide association study was conducted using a logistic regression model implemented in PLINK v.1.9 [29].

The logistic regression model was defined as:y = Xβ + Zu + e
where y is a $\ln=\frac{P}{1-P}$ with P being the probability of development of severe symptoms of COVID-19 infection; X and Z are design matrices, respectively, corresponding to β—a vector of non-genetic effects of age at the time of COVID-19 infection diagnosis and gender; u is an additive genetic effect of a variant; and e a residual. No population substructure has been observed in analysed data (Figure 3). The logistic regression model was also used for estimation of odds ratio (OR) along with untransformed standard error (SE) for each variant. We compared the distribution of observed *p*-values with expected *p*-values in the form of QQ-plots for both cardiovascular panel (Figure 4) and pulmonary panel (Figure 5). We defined two significance levels: less conservative Benjamin–Hochberg False Discovery Rate (FDR) < 0.1 and more conservative Bonferroni adjustment for α = 0.05.

## 5. Conclusions

In this study, we reported potential risk and protective variants in genes related to cardiovascular and pulmonary diseases that were related to a severe outcome of COVID-19 infection. Variants in the *HADHA* gene, described in this and previous studies, might play a significant role both as a protective and risk factor in the severe course of COVID-19. Most of the data in the COVID-19 cohorts are based on SNP microarrays and WES. Hence, our results stressed the importance of WGS-based analyses that complement and economically drive experiential designs using WES and reveal important aspects of the genetic background underlying severe COVID-19 infection. 

## Figures and Tables

**Figure 1 ijms-23-08696-f001:**
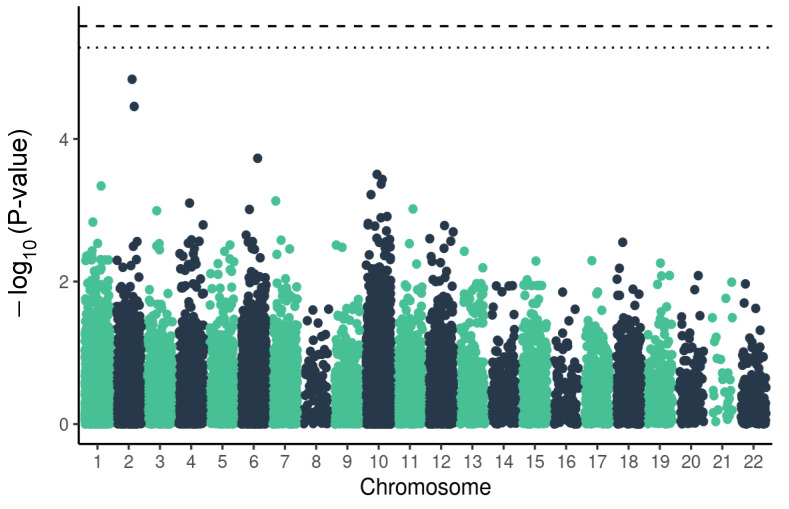
Manhattan plot of *p*-values for variants from the panel of genes associated with cardiovascular diseases and severe COVID-19 outcome. The dashed line marks the threshold for Bonferroni correction at α = 0.05, the dotted line marks FDR threshold < 0.1.

**Figure 2 ijms-23-08696-f002:**
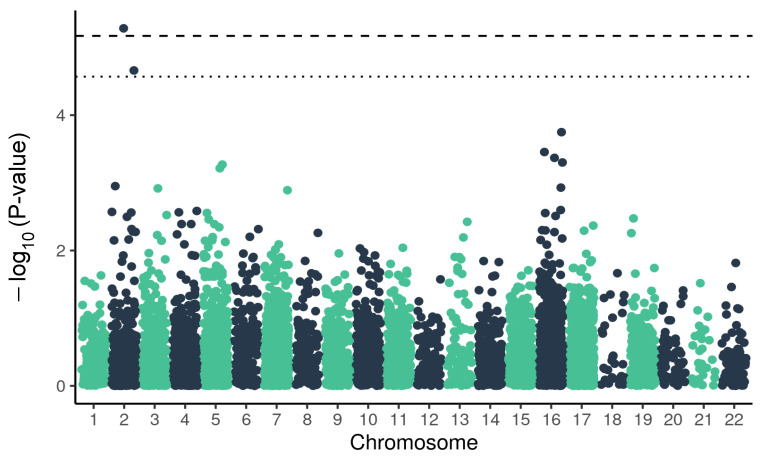
Manhattan plot of *p*-values for variants from the panel of genes associated with pulmonary diseases and associated with COVID-19 severity. The dashed line marks the threshold for Bonferroni correction at α = 0.05, the dotted line marks FDR threshold < 0.1.

**Figure 3 ijms-23-08696-f003:**
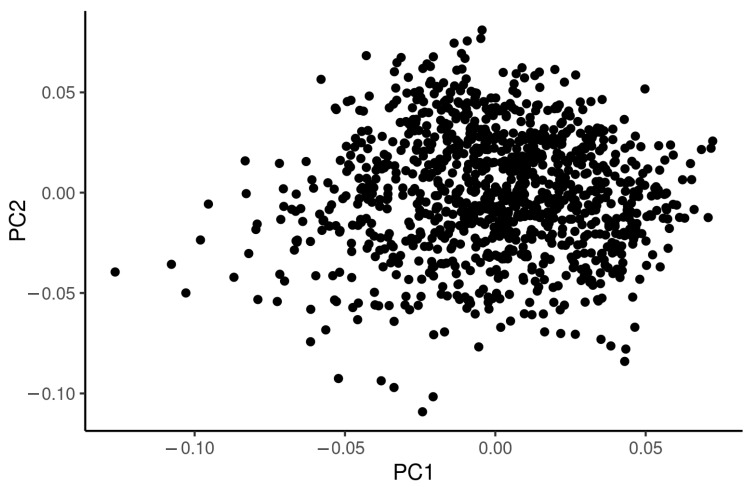
PCA plot of first two principal components of genomic relationship matrix for analysed variants.

**Figure 4 ijms-23-08696-f004:**
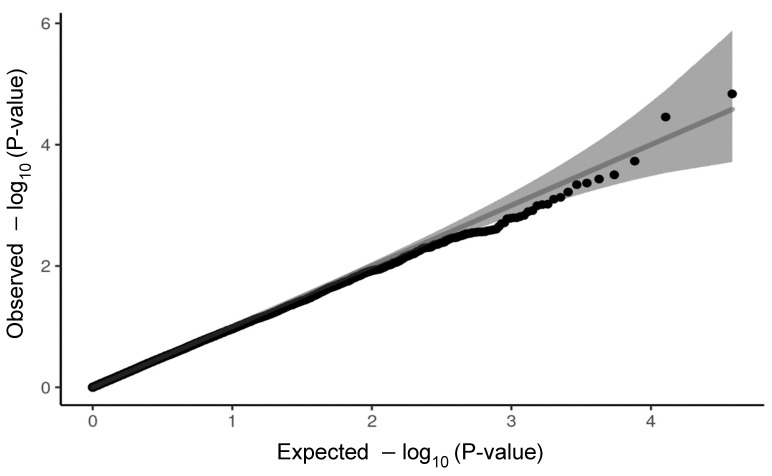
QQ-plot of *p*-values for cardiovascular panel. The gray shaded area indicates the 95% confidence interval under the null.

**Figure 5 ijms-23-08696-f005:**
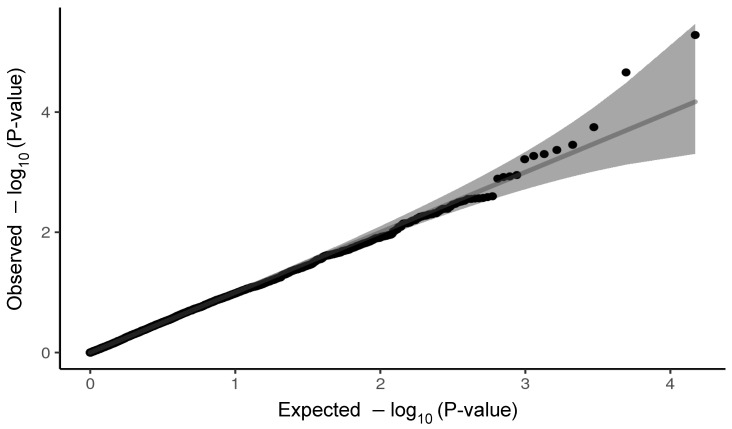
QQ-plot of *p*-values for pulmonary panel. The gray shaded area indicates the 95% confidence interval under the null.

**Table 1 ijms-23-08696-t001:** A variant with the lowest significance *p*-value from the panel of genes associated with cardiovascular diseases and related to COVID-19 severity, including its position, raw significance, Bonferroni-adjusted significance (BF) and FDR-adjusted significance (FDR), odds ratio (OR), untransformed standard error (SE) for odds ratio, and minor allele frequency (MAF).

Gene	Genomic Position (GRCh38)	Variant	Raw *p*-Value	BF	FDR	OR	SE	MAF
*HADHA*	chr2:26242159	rs56218721 G > C	0.000014	0.28	0.28	0.59	0.75	0.47

**Table 2 ijms-23-08696-t002:** Statistically significant variants associated with pulmonary diseases and related to COVID-19 severity, including its position, raw significance, Bonferroni-adjusted significance (BF) and FDR adjusted-significance (FDR), odds ratio (OR), untransformed standard error (SE) for odds ratio, and minor allele frequency (MAF).

Gene	Genomic Position (GRCh38)	Variant	Raw *p*-Value	BF	FDR	OR	SE	MAF
*DRC1*	chr2:26447115	rs10193369 A > T	0.0000052	0.038	0.039	0.56	0.13	0.48
*DRC1*	chr2:26414115	rs3067393 C > CATT	0.000022	0.16	0.08	1.71	0.13	0.43

## Data Availability

The dataset is freely available for researchers here https://1000polishgenomes.com/ (accessed on 1 August 2022) upon reasonable request [25].

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
