# Peer review of "Gene Variants Related to Cardiovascular and Pulmonary Diseases May Correlate with Severe Outcome of COVID-19"

_ijms, 2022, doi:10.3390/ijms23158696_

Round 1

Reviewer 1 Report

The authors report variants in genes related to cardiovascular and pulmonary diseases as risk-conferring and protective for a severe outcome of COVID-19 infection in a cohort of 1,076 unrelated Polish individuals. Importantly their work extends findings from previous association studies done using SNP arrays and WES.

The details of the results are cleared presented and the data will be of wide interest. The identification of multiple putative protective SNPs in LD is of particular merit.

Minor comments:

I believe the authors somewhat under-value the possible impact of intronic variants. While it is reasonable to propose that the identified variant mark a haplotype block, it is possible that the causative variant is intronic and excerpts its affect through gene regulation rather than a change in protein sequence. Therefore please consider rewording line 234 to:

causal mutations themselves. However, it is likely that their highly significant effects are due to strong LD with causal mutations that are more difficult to detect in cohorts of 235 a moderate size, both in GWAS and WES, due to a low frequency of the risk/protective 236 allele.

To aid readers, please also consider the following suggestions:

Suggested Language Edits:

I found the language of the abstract to be somewhat redundant (see lines 30 and 32 for example)lines.

Main text:

To avoid redundant language, please re-word lines 45 and 46 from:

Between the 1st of January 2020 and the 31st of December 2021, the pandemic of coronavirus disease 2019 (COVID-19) caused global excess mortality of 14.91 million people [1].

Line 49 to:

sonal risk factors of COVID-19 are age, male sex, and comorbidities [2–4], but these do not

Lin 61 to:

COVID-19 [7,8]. Studies on the influence of respiratory diseases, such as asthma and

Line 95:

potential severe COVID-19 risk variants. The highest impact was for a variant localised on

Line 97:

Please remove the bracket after the word genes.

Line 114:

Please delete the word ‘and’ at the start of the sentence and replace with a comma ‘,’

Line 149 to:

Previous studies indicated that CVDs were significant risk factors for a severe COVID-

Line 154 to:

PD. In our study, we searched for variants significantly associated with COVID-19 severity that have

Line 196 to:

studies, and one study reporting variants in CACNB2 as COVID risk variants associated with an

Line 199 to:

significantly associated with COVID-19 severe outcome, with 8 additional variants being in high LD. The

Line 201 to :

gene (OR = 13.52), and in the DNAH11 gene (OR = 11.08) (Table 3). Known pathogenic muta-

Line 226 to:

It has also been proposed as a potential prognostic marker for malignant mesothelioma [53,54].

Line 242 to: 

HADHA and HADHB were still significant. A previous study showed that mutations in

Author Response

To address all concerns and comments of both Reviewers, we re-analized our data. Unnecessary information was removed. It resulted in extensive changes in the manuscript, therefore we did not use tracked changes mode to mark all of them. New figures and tables were prepared. Previous tables 3 and 4 were removed according to the Reviewer’s #2 suggestions. Detailed answers to the Reviewers’ comments and description of changes done are given below. We have also thanked the Reviewers in the Acknowledgements section.

Reviewer #1

The authors report variants in genes related to cardiovascular and pulmonary diseases as risk-conferring and protective for a severe outcome of COVID-19 infection in a cohort of 1,076 unrelated Polish individuals. Importantly their work extends findings from previous association studies done using SNP arrays and WES. The details of the results are cleared presented and the data will be of wide interest. The identification of multiple putative protective SNPs in LD is of particular merit.

Answer: Thank you for this comment.

Minor comments:

I believe the authors somewhat under-value the possible impact of intronic variants. While it is reasonable to propose that the identified variant mark a haplotype block, it is possible that the causative variant is intronic and excerpts its affect through gene regulation rather than a change in protein sequence. Therefore please consider rewording line 234 to:

causal mutations themselves. However, it is likely that their highly significant effects are due to strong LD with causal mutations that are more difficult to detect in cohorts of 235 a moderate size, both in GWAS and WES, due to a low frequency of the risk/protective 236 allele.

Answer: Done.

I found the language of the abstract to be somewhat redundant (see lines 30 and 32 for example) lines.

Answer: Abstract was written again, according to the Reviewer’s suggestions.

To avoid redundant language, please re-word lines 45 and 46 from: Between the 1st of January 2020 and the 31st of December 2021, the pandemic of coronavirus disease 2019 (COVID-19) caused global excess mortality of 14.91 million people [1].

Answer: Done.

Line 49 to: sonal risk factors of COVID-19 are age, male sex, and comorbidities [2–4], but these do not

Answer: Done.

Lin 61 to: COVID-19 [7,8]. Studies on the influence of respiratory diseases, such as asthma and

Answer: Done.

Line 95: potential severe COVID-19 risk variants. The highest impact was for a variant localised on

Answer: Done.

Line 97: Please remove the bracket after the word genes.

Answer: Done.

Line 114: Please delete the word ‘and’ at the start of the sentence and replace with a comma ‘,’

Answer: This fragment was removed.

Line 149 to: Previous studies indicated that CVDs were significant risk factors for a severe COVID-

Answer: Done.

Line 154 to: PD. In our study, we searched for variants significantly associated with COVID-19 severity that have

Answer: Done.

Line 196 to: studies, and one study reporting variants in CACNB2 as COVID risk variants associated with an

Answer: This fragment was removed.

Line 199 to: significantly associated with COVID-19 severe outcome, with 8 additional variants being in high LD. The

Answer: This fragment was removed.

Line 201 to: gene (OR = 13.52), and in the DNAH11 gene (OR = 11.08) (Table 3). Known pathogenic muta-

Answer: This fragment was removed.

Line 226 to: It has also been proposed as a potential prognostic marker for malignant mesothelioma [53,54].

Answer: This fragment was removed.

Line 242 to: HADHA and HADHB were still significant. A previous study showed that mutations in

Answer: Done.

Reviewer 2 Report

Major:

A demographic summary table of the subjects included in the analysis should be provided.  The authors cite a previous paper, but this is insufficient.  Similarly, the design of the study sample collection should

It’s unclear to me what the rationale is for including uninfected individuals is in the control group for the GWAS analysis.  Would this not attenuate any true associations with disease severity since the subjects in question are not exposed, as well as potentially yield false positive associations of genetic variants that confer resistance to infection?  If the target phenotype of interest is severity, than this should probably be treated as a multi-class outcome with uninfected as a separate class, or they should be excluded from consideration.  Similarly, as per the previous comment, the number of individuals in each of these classes should be summarized.

The authors prune the genetic variation for purposes of GWAS, although the rationale for this is unclear.  Why not just tests all variants?  This is kind of backwards relative to how genetic analysis has progressed (where tag SNPs were selected due technical/cost constraints).

I don’t understand the rationale for defining the significance threshold post-hoc per the QQ plot.  This should be based on the effective number of independent variants (which can be readily estimated using standard methods), not based on the results of the analysis.

The analytical methodology is fairly simple relative to the proposed hypothesis (that genetic variants RELATED to cardiovascular and/or pulmonary diseases) are similarly associated with COVID-19 severity.  By simply focusing on genes, this discards information regarding previously identified associations (and directions of effect) and the single variant level, using methods like colocalization, genetic correlation, and possibly mendelian randomization.

It seems like a waste to ignore potentially important variation outside yet proximal to the genes.  Did the authors consider buffer regions about these genes to capture potential regulatory variants?  To this effect, the filtering per gene lists is not really clear vis a vis positional information.  Is this just start and stop of the CDS?

Lack of validation cohort.  There are growing numbers of COVID-19 severity GWAS; is there evidence of replicability of any of these findings elsewhere?

Minor: 

The GWAS included covariate adjustment for age and sex, but was there any evidence of population substructure that should be accounted for?  The cohort description sounds relatively homongeneous and the QQ plots don’t demonstrate any substantial deviation that would indicate uncorrected population stratification, but this should be mentioned somewhere.

Variant association summary tables should also include MAF.

Lines 191-192:  Sentence ends abrupty (“Two other protective variants in a genomic region that correlated with significant 191 variants were found in the”).

How was HWE testing performed?  If you have cases and controls, you should restrict this to the controls.

“More importantly, after selecting even more restrictive p-value threshold, variants in 241 HADHA and HADHB were still significant. “  I’m not sure what this means.  As noted above, thresholds should be defined a priori per the analysis design, and variants are either significant or not.

I don’t really see too much utility in Table 2, aren’t these just reflective of the same significant loci in table 1?

Effect size interpretation:  The authors describe ORs from to 2-8 as “low”, which doesn’t seem low to me (these are very large genetic effects).  Also, the extremity of the estimate for rs1019502135 is likely highly driven by low MAF (did this pass the 1% MAF threshold?), and also has a similarly extreme degree of uncertainty (per reported CIs). 

Discretionary:

I generally find reporting of 95% CI’s for high-dimensional discovery analyses not very useful (and can be misleading), since it ignores the extreme multiple testing burden.  Reporting SE’s (untransformed or transformed) alongside the OR estimates is more appropriate, but technically you can get that from the CI’s.

Author Response

To address all concerns and comments of both Reviewers, we re-analized our data. Unnecessary information was removed. It resulted in extensive changes in the manuscript, therefore we did not use tracked changes mode to mark all of them. New figures and tables were prepared. Previous tables 3 and 4 were removed according to the Reviewer’s #2 suggestions. Detailed answers to the Reviewers’ comments and description of changes done are given below. We have also thanked the Reviewers in the Acknowledgements section.

Reviewer #2

A demographic summary table of the subjects included in the analysis should be provided.  The authors cite a previous paper, but this is insufficient.  Similarly, the design of the study sample collection should

Answer: Done. Additional Supplementary Materials 1 was added to the manuscript.

It’s unclear to me what the rationale is for including uninfected individuals in the control group for the GWAS analysis.  Would this not attenuate any true associations with disease severity since the subjects in question are not exposed, as well as potentially yield false positive associations of genetic variants that confer resistance to infection?  If the target phenotype of interest is severity, then this should probably be treated as a multi-class outcome with uninfected as a separate class, or they should be excluded from consideration.  Similarly, as per the previous comment, the number of individuals in each of these classes should be summarized.

Answer: Following the reviewer’s suggestion we decided to exclude uninfected patients from the control group, leaving only mild and asymptomatic patients.

The authors prune the genetic variation for purposes of GWAS, although the rationale for this is unclear.  Why not just test all variants?  This is kind of backwards relative to how genetic analysis has progressed (where tag SNPs were selected due technical/cost constraints).

Answer: Due to the small cohort in terms of association studies (<1000 individuals) we decided to reduce the number of tested variants in order to minimize risk of type I errors.

I don’t understand the rationale for defining the significance threshold post-hoc per the QQ plot.  This should be based on the effective number of independent variants (which can be readily estimated using standard methods), not based on the results of the analysis.

Answer: Following the reviewer’s suggestion we decided to apply two significance thresholds: less conservative Benjamin-Hochberg FDR < 0.1 and more conservative Bonferroni adjustment for α = 0.05.

The analytical methodology is fairly simple relative to the proposed hypothesis (that genetic variants RELATED to cardiovascular and/or pulmonary diseases) are similarly associated with COVID-19 severity.  By simply focusing on genes, this discards information regarding previously identified associations (and directions of effect) and the single variant level, using methods like colocalization, genetic correlation, and possibly mendelian randomization.

Answer: Thank you for this suggestion. In our study we decided to check if some genes associated with cardiovascular/pulmonary disorders are more associated with COVID-19 severity than others. At the moment of submitting this paper we do not have sufficient clinical data for study of genetic correlations or mendelian randomization.

It seems like a waste to ignore potentially important variation outside yet proximal to the genes.  Did the authors consider buffer regions about these genes to capture potential regulatory variants?  To this effect, the filtering per gene lists is not really clear vis a vis positional information.  Is this just start and stop of the CDS?

Answer: Following the reviewer’s suggestion we decided to add 10 kb buffer regions at both ends of every gene.

Lack of validation cohort.  There are growing numbers of COVID-19 severity GWAS; is there evidence of replicability of any of these findings elsewhere?

Answer: Unfortunately, none of the recent studies corroborate our results. Single case study of Wongkittichote (https://doi.org/10.1002/jmd2.12165) indicates HADHA genes as potential markers for severe COVID-19. To the best of our knowledge, DRC1 has not been associated with COVID-19 yet.  

Minor: 

The GWAS included covariate adjustment for age and sex, but was there any evidence of population substructure that should be accounted for?  The cohort description sounds relatively homogeneous and the QQ plots don’t demonstrate any substantial deviation that would indicate uncorrected population stratification, but this should be mentioned somewhere.

Answer: There was no population structure observed in both panels. We added a new Figure 3 with the first two principal components from the genomic relationship matrix to show it.

Variant association summary tables should also include MAF.

Answer: MAFs have been added to summary tables.

Lines 191-192:  Sentence ends abrupty (“Two other protective variants in a genomic region that correlated with significant 191 variants were found in the”).

Answer: Corrected.

How was HWE testing performed?  If you have cases and controls, you should restrict this to the controls.

Answer: Testing was performed for controls only with HWE exact test.

“More importantly, after selecting an even more restrictive p-value threshold, variants in HADHA and HADHB were still significant. “  I’m not sure what this means.  As noted above, thresholds should be defined a priori per the analysis design, and variants are either significant or not.

Answer: This fragment was removed.

I don’t really see too much utility in Table 2, aren’t these just reflective of the same significant loci in table 1?

Answer: Table 2 and Table 4 were removed after re-runing analysis.

Effect size interpretation:  The authors describe ORs from to 2-8 as “low”, which doesn’t seem low to me (these are very large genetic effects).  Also, the extremity of the estimate for rs1019502135 is likely highly driven by low MAF (did this pass the 1% MAF threshold?), and also has a similarly extreme degree of uncertainty (per reported CIs). 

Answer: The term “low” was removed when used along with OR. Due to re-analysis of data, this variant is no longer significant and was removed from the manuscript.

I generally find reporting of 95% CI’s for high-dimensional discovery analyses not very useful (and can be misleading), since it ignores the extreme multiple testing burden.  Reporting SE’s (untransformed or transformed) alongside the OR estimates is more appropriate, but technically you can get that from the CI’s.

Answer: Confidence intervals have been replaced with untransformed standard errors.

Round 2

Reviewer 2 Report

Major:

Some of the results still feel overstated in the discussion/conclusion.

Discussion:  “Moreover, HADHA and DRC1 genes described in this study were also identified as significant in case of different immunological responses to 170 COVID-19 (severe outcome or resistance to the disease) (in prep.).”  I don’t think you can cite something that is not peer reviewed as supporting evidence.

Conclusion:  “Although all identified SNPs in the WGS sequenced cohort were localised in introns, they are associated with causal variants and thus provide markers of a severe outcome of COVID-19 infection”  I don’t know what this means – “causal” how?  Based on what?

Conclusion: “Variants in HADHA gene, described in this and previous studies, deserve the most attention” This feels overstated given the association does not meet even the more liberal significance thresholds outlined by the authors.

Minor:

PC plots:   It seems like these were generated from the SNPs selected from the panels (and thus have two different sets of results for the same set of subjects.  That doesn’t make sense – just use the full data available.

Introduction:  The authors indicate they are testing whether genetic variants that are previously associated with CVD and PD, are associated with COVID.  In reality, the are testing variants located in genes which are associated with CVD.  This is a subtle but important difference.

Introduction:  “223 COVID-19 patients”  Not sure what the 223 denotes.

Discussion:  “Due to their genomic location in non-coding regions, these variants are not expected to be causal mutations themselves. However, it is likely that their highly significant effects are due to strong LD with causal mutations that are more difficult to detect in cohorts of a moderate size, both in GWAS and WES, due to a low frequency of the risk/protective allele. “  I’m not sure what the authors are trying to say here – the variants they identified were all common, and thus any other variants in high LD would also likely be common (not rare).  The authors could obviously pursue statistical finemapping if they wanted, given they have NGS data available.

Results:  “The logistic regression GWAS analysis of 19,130 SNPs and indels located in genes 92 related to CVD identified 1 significant variant in COVID-19 severe outcome, in Hydroxy- 93 acyl-CoA Dehydrogenase Trifunctional Multienzyme Complex Subunit Alpha (HADHA) 94 gene (OR = 0.59)”  Please also add p-value to accompany OR estimate (here and elsewhere).  Also, this isn’t significant per the definitions outlined by the authors.

Table 1:  “A variant with the lowest significance level”. This is not what significance level means.  A significance level is the alpha level, which is the same thing as the statistical significance threshold.  This is the variant with the lowest p-value.

Results: “in in” typo (line 108).

Author Response

Discussion:  “Moreover, HADHA and DRC1 genes described in this study were also identified as significant in case of different immunological responses to 170 COVID-19 (severe outcome or resistance to the disease) (in prep.).”  I don’t think you can cite something that is not peer reviewed as supporting evidence.

Answer: It was removed.

Conclusion:  “Although all identified SNPs in the WGS sequenced cohort were localised in introns, they are associated with causal variants and thus provide markers of a severe outcome of COVID-19 infection”  I don’t know what this means – “causal” how?  Based on what?

Answer: It was removed

Conclusion: “Variants in HADHA gene, described in this and previous studies, deserve the most attention” This feels overstated given the association does not meet even the more liberal significance thresholds outlined by the authors.

Answer: It was removed.

PC plots:   It seems like these were generated from the SNPs selected from the panels (and thus have two different sets of results for the same set of subjects.  That doesn’t make sense – just use the full data available.

Answer: PC plots are replaced with single plot for all analysed data.

Introduction:  The authors indicate they are testing whether genetic variants that are previously associated with CVD and PD, are associated with COVID.  In reality, the are testing variants located in genes which are associated with CVD.  This is a subtle but important difference.

Answer: Changed according to the Reviewer’s suggestion.

Introduction:  “223 COVID-19 patients”  Not sure what the 223 denotes.

Answer: It was changed.

Discussion:  “Due to their genomic location in non-coding regions, these variants are not expected to be causal mutations themselves. However, it is likely that their highly significant effects are due to strong LD with causal mutations that are more difficult to detect in cohorts of a moderate size, both in GWAS and WES, due to a low frequency of the risk/protective allele. “  I’m not sure what the authors are trying to say here – the variants they identified were all common, and thus any other variants in high LD would also likely be common (not rare).  The authors could obviously pursue statistical finemapping if they wanted, given they have NGS data available.

Answer: It was removed

Results:  “The logistic regression GWAS analysis of 19,130 SNPs and indels located in genes 92 related to CVD identified 1 significant variant in COVID-19 severe outcome, in Hydroxy- 93 acyl-CoA Dehydrogenase Trifunctional Multienzyme Complex Subunit Alpha (HADHA) 94 gene (OR = 0.59)”  Please also add p-value to accompany OR estimate (here and elsewhere).  Also, this isn’t significant per the definitions outlined by the authors.

Answer: Corrected

Table 1:  “A variant with the lowest significance level”. This is not what significance level means.  A significance level is the alpha level, which is the same thing as the statistical significance threshold.  This is the variant with the lowest p-value.

Answer: Corrected

Results: “in in” typo (line 108).

Answer: Corrected.